

# Overcoming pain thresholds with multilevel models—an example using quantitative sensory testing (QST) data

Gerrit Hirschfeld[1], Markus R. Blankenburg[2,3], Moritz Süß[4] and Boris Zernikow[2,5]

[1] Faculty of Business Management and Social Sciences, University of Applied Sciences Osnabrück, Osnabrück, Germany
[2] Chair for Children's Pain Therapy and Paediatric Palliative Care, Witten/Herdecke University, Witten/Herdecke, Germany
[3] Department for Pediatric Neurology, Psychosomatic and Pain Medicine, Center for Child and Adolescent Medicine Olgahospital, Klinikum Stuttgart, Stuttgart, Germany
[4] Department for Psychology, University Düsseldorf, Düsseldorf, Germany
[5] German Pediatric Pain Centre—Children's Hospital Datteln, Witten/Herdecke University, Germany

Corresponding author
Gerrit Hirschfeld,
hirschfeld@wi.hs-osnabrueck.de

## ABSTRACT

The assessment of somatosensory function is a cornerstone of research and clinical practice in neurology. Recent initiatives have developed novel protocols for quantitative sensory testing (QST). Application of these methods led to intriguing findings, such as the presence lower pain-thresholds in healthy children compared to healthy adolescents. In this article, we (re-) introduce the basic concepts of signal detection theory (SDT) as a method to investigate such differences in somatosensory function in detail. SDT describes participants' responses according to two parameters, sensitivity and response-bias. Sensitivity refers to individuals' ability to discriminate between painful and non-painful stimulations. Response-bias refers to individuals' criterion for giving a "painful" response. We describe how multilevel models can be used to estimate these parameters and to overcome central critiques of these methods. To provide an example we apply these methods to data from the mechanical pain sensitivity test of the QST protocol. The results show that adolescents are more sensitive to mechanical pain and contradict the idea that younger children simply use more lenient criteria to report pain. Overall, we hope that the wider use of multilevel modeling to describe somatosensory functioning may advance neurology research and practice.

# INTRODUCTION

The assessment of somatosensory function is a cornerstone of research and clinical practice in neurology. Understanding both acute as well as chronic pain relies, to a large extent, on our ability to measure and quantify the response to painful and non-painful stimuli. Many resources have been used to develop novel and standardized methods to collect such data, e.g., the quantitative sensory testing (QST) protocol of the German research network on
neuropathic pain (DFNS) (*Rolke et al., 2006*). The QST protocol has been adapted for other languages (*Schestatsky et al., 2011*) and children (*Blankenburg et al., 2010*). The availability of such standardized measures has resulted in a rapid increase in the number of studies into pain processing in healthy participants and patients. These experimental paradigms have also been used to investigate long-standing observations in clinical practice, e.g., higher pain-ratings in women compared to men (*Fillingim et al., 2009*) or higher pain-ratings in girls compared to boys (*Goodenough et al., 1999*). While some of these clinical findings can be replicated in experimental paradigms, researchers have also identified some differences. In pediatric settings, girls report higher pain-intensity ratings following venipuncture than boys (*Goodenough et al., 1999*), while such differences are not apparent in QST data (*Blankenburg et al., 2011*). This may be because experimental studies describe participants' responses exclusively in terms of thresholds that conflate several aspects of the response. One method to tease out several aspects of participants' responses is signal detection theory (SDT). While SDT has been criticized in the past (*Clark, 1974*; *Rollman, 1976*; *Chapman, 1977*; *Gracely, 2006*), we believe that several recent developments concerning the standardization of QST protocols and data-analysis should lead to a reassessment of SDT.

The aim of the present paper is to re-introduce the basic concepts of (SDT) and show how modern multilevel models can be used to estimate the two SDT parameters, sensitivity and response bias, from the mechanical pain sensitivity test within the QST protocol of the DFNS.

## Thresholds

The most intuitive and widely used way to describe participants' pain reports is in terms of thresholds (*Gracely, 2006*). Pain thresholds are typically defined as the stimulus intensity at which participants report experiencing pain in 50 percent of the trials or the intensity at which animals show a withdrawal response in 50 percent of the trials (*Mills et al., 2012*). There are two different methods to determine pain thresholds; the *methods of constant stimuli* (also known as *the method of levels)* and the *method of limits*. When the *method of constant stimuli* is used, participants are presented with stimuli that are above and below the pain-threshold. After each stimulus participants report whether the stimulus was painful or not. The threshold is determined as the stimulus intensity at which 50% of the stimuli are rated as painful. The main disadvantage of this method is that it takes some time to present all different stimulus intensities. This may result in fatigue and sensitization that both affect the measurement. That is why most stimulation protocols for clinical application use the more time-efficient *method of limits,* i.e., alternating between a gradually increasing or decreasing series of stimulus intensities until a participant changes the response from non-painful to painful, or vice versa. The threshold is determined as the stimulus intensity at which participants change their response. The problems with the *method of limits* becomes apparent when it is applied to other domains of sensory testing, e.g., visual acuity testing (Fig. 1). Because it is very easy for participants to predict upcoming stimuli, they can easily fake their test results. Furthermore, when the intensity of stimuli can only by gradually increased, e.g., heat when determining heat-pain thresholds,

**A**

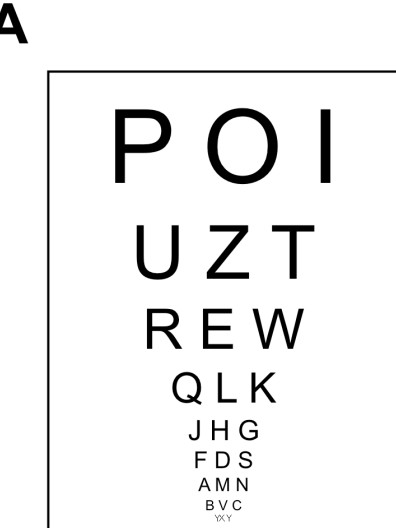

*Read aloud from
top to bottom.*

**B**

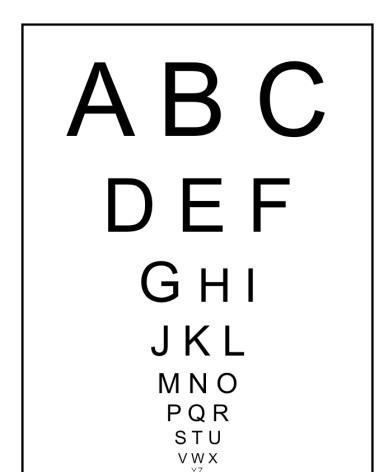

*Read aloud from
top to bottom.*

**Figure 1 Problems associated with the method of limits.** (A) Analogue to the method of constant stimuli: randomized order of stimuli, as in the method of constant stimuli, prohibits any predictions. (B) Analogue to the method of limits: ordered presentation makes predicting the next stimulus extremely easy.

participants' reaction time will affect the thresholds. Specifically, when the heat is increased until participants report a painful response, participants with fast reaction times will appear to have lower heat-pain thresholds than participants with slow reaction times.

## Signal detection theory

An alternative to describing participants' responses in terms of thresholds is to use signal detection theory to model their responses within individual trials (*Green & Swets, 1966*; *Macmillan & Creelman, 2004*). However, to use this method, data need to be collected using the *method of constant stimuli* (gathering individual ratings on a random series of stimuli, including empty trials without stimulation). According to signal detection theory, participants' responses can be modeled by two parameters, sensitivity and response bias. Sensitivity refers to the ability to accurately discriminate between the presence and absence of a target stimulus. Response bias refers to participants' criteria for reporting the presence or absence of the stimulus. Differences in these parameters become apparent when stimulus-response-functions are plotted for a number of participants (see Fig. 2 below). Participants with high sensitivity show a steep increase in the percent of painful responses with increasing stimulus intensities. Participants with high criteria for pain show these increases at higher stimulus intensities. The most widely used metrics for sensitivity and response bias are $d'$ and $c$, respectively. These two parameters are readily calculated from a $2 \times 2$ table consisting of the counts for hits (participants correctly report presence

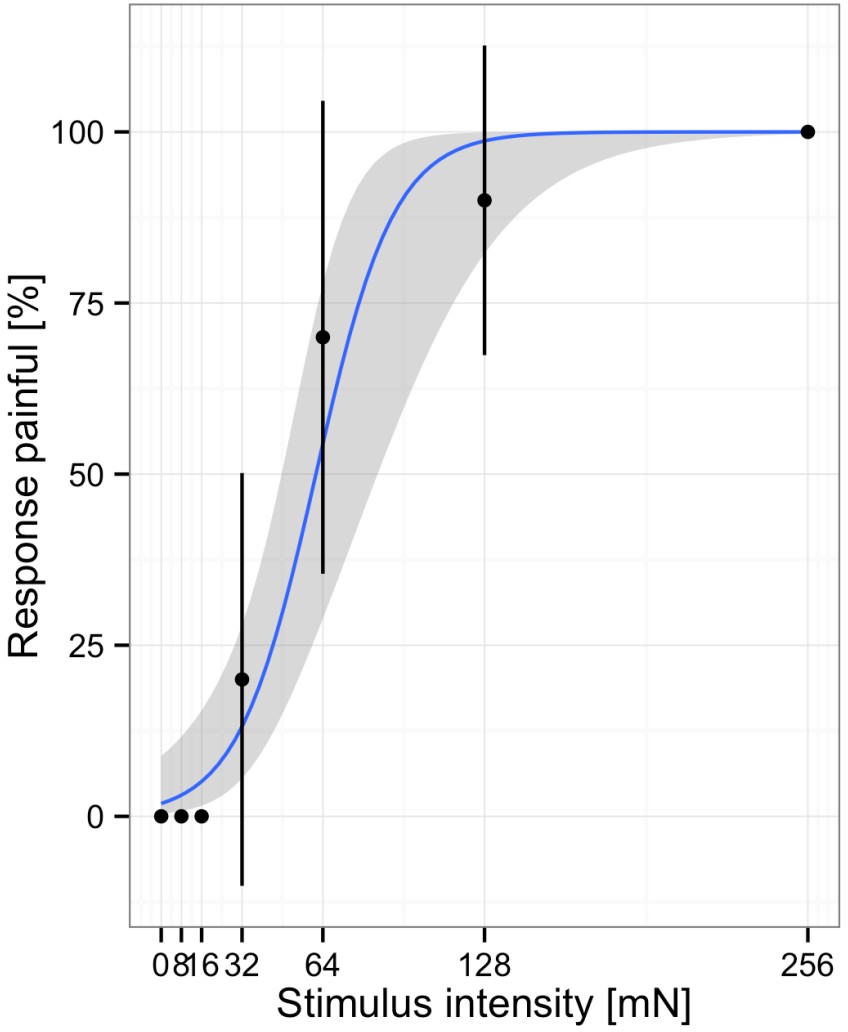

**Figure 2 Responses and fitted model for one subject.** Participants rated stimuli at different intensities. Note, points represent the average % of responses painful at each stimulus intensity, the blue line indicates the fit to the individual data.

of stimuli), misses (participants report absence of stimuli even though they are present), false-alarms (participants report presence of stimuli even though they are absent), and correct rejections (participants correctly report absence of stimuli) (*Green & Swets, 1966*). From these the hit rate (hits/(hits + misses)) and false-alarm rate (false-alarms/(false alarms + correct rejections)) are computed. The measure $d'$ is defined as the $z$-score corresponding to the hit rate minus the $z$-score corresponding to the false-alarm rate. If participants are very good at distinguishing the presence or absence of stimuli, they have many hits and correct rejections and few false-alarms and misses resulting in high values for $d'$. If participants are bad at distinguishing the presence or absence of stimuli and perform at chance level, $d'$ will be zero. The measure $c$ is defined as $-.5$ (-score corresponding to the hit rate plus the $z$-score corresponding to the false-alarm rate). If participants show no preference for either response, $c$ is zero. If participants have

a preference towards a specific response this will be either positive or negative. More information on the calculation and interpretation of SDT measures is given by *Stanislaw & Todorov (1999)*.

Importantly, it is much more difficult for participants to willingly change the test-results into a specific direction. First, because the SDT measures $d'$ and $c$ are much less transparent to participants than thresholds determined using the method of limits. With the latter most participants will understand that they only have to report "painful" to light touch to fake sings of a painful condition. However, only very few will have a profound-enough understanding of SDT to change the sensitivity to pain as determined by SDT towards a specific direction. Second, because in analogy to visual acuity testing, it is impossible to correctly identify the letters if they are not discernable. That is it is impossible to fake being able to discern specific stimuli isf in reality one cannot do so.

Crawford Clark was among the first to apply the SDT model to pain assessment (*Clark, 1974*). This initiated a highly active and controversial discussion about the application of psychophysical methods to pain research in the 1970s and 1980s (*Clark, 1974*; *Rollman, 1976*; *Chapman, 1977*; *Gracely, 2006*). Clark's contention that sensitivity is only affected by "neurologic" factors, while response bias is only affected by "psychologic" factors, made this purely descriptive approach at the same time highly interesting and controversial. Using this dichotomy, he demonstrated that interventions, such as acupuncture, only affect response bias but not sensitivity (*Clark & Yang, 1974*). Similarly, he found that chronic pain patients differ from healthy controls in that they use more stoical criteria for painful stimuli (*Yang et al., 1985*). There were, however, many criticisms for this approach and especially Clark's assumption that the two SDT-parameters are influenced only by the supposed factors (*Rollman, 1979*). Ultimately, this assumption did not stand up to further empirical tests. For example, Gracely and colleagues showed that "psychologic" factors may affect sensitivity (*Gracely, 2006*). However, SDT is still a useful framework for describing participants' performance, even though Clark's strict interpretation of these parameters as indexes for "psychologic" and "neurologic" factors is likely wrong.

Additionally, two technical problems averted the wider use of these techniques. First, there were no established protocols that would specify the mode of stimulation and intensity. As has been noted before, using different methods results in incomparable findings across studies (*Rollman, 1976*). However, the introduction of the DFNS-QST protocol has homogenized the methods to collect data that can be analyzed using SDT. Specifically, the mechanical pain sensitivity test lends itself easily for an analysis in terms of SDT (*Rolke et al., 2006*). Second, if the SDT parameters are estimated via $2 \times 2$ tables as introduced by Clark and colleagues, only binary responses and two levels of stimuli can be processed. That is, even if several stimulus intensities are used, these have to be dichotomized into painful and non-painful stimuli, resulting in a loss of variance and possibly different effects, depending on the chosen cutpoint (*Rollman, 1976*). However, several recent extensions of multilevel models (*Wright, Horry & Skagerberg, 2009*; *Wright & London, 2009*) make it possible to estimate the SDT-parameters using the full information conveyed in participants' responses.

## Multilevel models

It has been known for some time that generalized linear models (GLMs) can be used to estimate the SDT-parameters (*DeCarlo, 1998*). Namely, the slope and intercept of a probit-regression function are analogues to the $d'$ and $c$ parameters estimated by traditional SDT. As long as the stimulus and response variables are dichotomized, these will provide the numerically same results as the approach based in $2 \times 2$ tables (*Wright & London, 2009*). The biggest advantage is, however, that multilevel models can also address continuous stimuli and responses. Specifically, using continuous measures for stimulation circumvents the above-mentioned problem that there are no objective criteria to judge whether pain is present or absent. Furthermore, if these models are fit in a multilevel context, these techniques are much more flexible in terms of the collected data and group-differences that can be tested.

While such methods were only previously available in highly specialized software, there are a number of implementations to fit GLMs using the open-source software R (*R Core Team, 2012*). Currently, the most widely used implementation to fit GLMs is the lme4 package (*Bates et al., 2014*) and specialized packages that access functions from the LME4 package to fit psychophysical models (*Wright, Horry & Skagerberg, 2009*).

## EXAMPLE APPLICATION: AGE AND GENDER EFFECTS IN MECHANICAL PAIN SENSITIVITY

### Methods

In order to examine the effects of age and gender on mechanical pain sensitivity we re-analyzed data from 172 healthy children ($n = 85$; 41 female) and adolescents ($n = 87$; 41 female) who took part in an earlier study (Fig. 2) (*Blankenburg et al., 2011*). The study was approved by the Ethics Committee of the Witten/Herdecke University (101/2008) and encompassed written informed consent from children and their guardians.

Participants completed the QST protocol of the German Research Network on Neuropathic Pain (*Rolke et al., 2006*) adopted for children (*Blankenburg et al., 2010*). The QST protocol consists of thirteen tests that assess both nociceptive as well as non-nociceptive modalities and afferent nerve fibers and central pathways. The whole protocol is widely used to study both healthy subjects and patients (*Magerl et al., 2010*; *Maier et al., 2010*; *Mücke et al., 2014*) as well as children (*Blankenburg et al., 2011*; *Hirschfeld et al., 2012*). Here we analyzed the data from the Mechanical Pain Sensitivity (MPS) test. Within the MPS six different pinprick mechanical stimulators with weights between 8 mN and 256 mN, and three light tactile stimulators (cotton wisp, cotton wool tip and brush) were applied to the back of both hands in pseudorandomized sequence. Each of the nine stimulators was applied ten times (five times on each hand) resulting in 90 ratings for each child and a total of 15.480 ratings for all children.

The analysis proceeded in two steps: model building and assessment. First, we specified a baseline model that predicted the response (painful or not painful) using only a constant as a fixed effect and *participants* and *stimulus intensity* as random effects (*Barr et al., 2013*). The random effect for *participant* indexes the individual intercept that can be interpreted

**Table 1** Important functions to fit GLMs to psychophysical data.

| Function | What it does |
| --- | --- |
| *glmer()* | Fits a model to data. The example code specifies a dependent variable "res_01," fixed effects "Stimulus," random effects "(Stimulus\|Code)," a link function "binomial(link = logit)," and the data to which this is fitted. |
| | Example: |
| | >mod_01<-glmer(res_01 ~ Stimulus + (Stimulus\|Code), family = binomial(link = logit), data = data) |
| *anova()* | Compares fitted models to each other. The example code compares three consecutively more complex models. |
| | Example: |
| | >anova(mod_01, mod_02, mod_03) |
| *summary()* | Gives an overview of the model, including parameter estimates and significance levels. |
| | Example: |
| | >summary(mod_03) |
| *ranef() and fixef()* | Prints the random and fixed effects. |
| | Example: |
| | >ranef(mod_3) |
| | >fixef(mod3) |

in terms of response bias with high values indicating that participants were more likely to report a painful response. The random effect for *stimulus intensity* indexes the individual slope parameter and may be interpreted in terms of the sensitivity. Specifically, high values on the slope parameter indicate that participants show a steep increase in painful responses with increasing stimulus intensity. Conversely, low values indicate that participants show only a weak increase in painful responses with increasing stimulus intensity. We then sequentially added *stimulus intensity, age* and *sex* as fixed effects. These fixed effects describe effects that hold for the whole group of participants, i.e., the fixed effect for *stimulus intensity* describes that across all participants high stimulus intensities were more likely to elicit a "painful" response. For each effect that was added we checked whether the inclusion significantly improved the model fit (*Baayen, Davidson & Bates, 2008*). Only effects that resulted in a significant improve the model fit were retained in the final model. Second, we inspected the final model and extracted information on the fixed effects, because these describe relevant effects at the group level. These steps are readily implemented in R (see Table 1 for an overview of relevant functions).

## Results

In a first step the responses by individual participants were visualized to inspect whether the responses conform to the assumed model or not. As can be seen in Figs. 2 and 3 the model fitted the responses of individual participants pretty well.

To arrive at a suitable final model, a series of models were fit and compared to each other (Table 2). Adding *stimulus intensity* as a fixed effect to the baseline model greatly improved the model fit ($\chi^2 = 94.7$; $df = 1$; $p < .001$). Adding *sex* and the interaction between *sex*

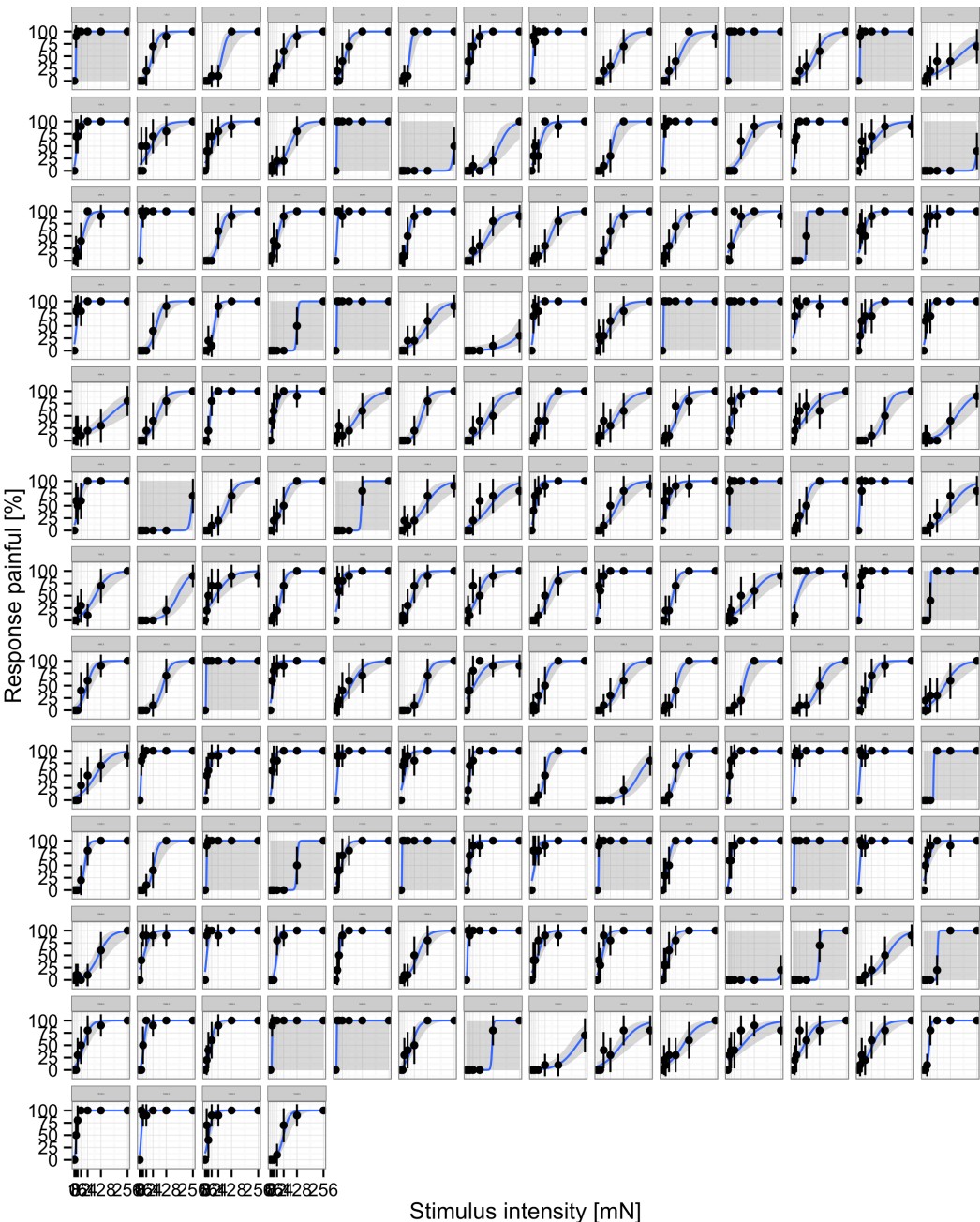

**Figure 3 Responses of individual participants.** Each panel represents an individual, points represent the average % of responses painful at each stimulus intensity, the blue line indicates the fit to the individual data.

and *stimulus intensity* as a fixed effect to this model did not result in a significant increase in model fit ($\chi^2 = 1.79$; df $= 2$; $p < .41$); therefore, these effects were not added to the model. This indicates that there were no systematic differences in sensitivity and response bias between boys and girls. Adding *age* and the interaction between *age* and *stimulus intensity*

**Table 2  Series of model comparisons.**

| Model | $\chi^2$ (diff $\chi^2$) | DF (diff DF) | $p$ ($\chi^2$) |
|---|---|---|---|
| M1: baseline | | | |
| M2: model M1 plus *stimulus* | (94.7011) | 1 | <.001 |
| M3: model M2 plus *sex and stimulus * sex* | (1.7872) | 2 | 0.41 |
| M4: model M2 plus *age and stimulus * age* | (8.847) | 2 | .01 |

**Table 3  Estimates of the final model (M4).**

| Parameter | Estimate | SE | OR | Z | $p$ |
|---|---|---|---|---|---|
| Intercept | −3.29 | .14 | 0.04 | −23.11 | <.001 |
| Stimulus | 0.10 | .02 | 1.11 | 6.20 | <.001 |
| Age (children) | 0.14 | .20 | 1.15 | .72 | .473 |
| Stimulus * Age | .07 | .02 | 1.07 | 2.97 | .003 |

**Notes.**
Reference category in parenthesis.

to the model significantly improved the model fit ($\chi^2 = 8.85$; df $= 2$; $p = .01$), indicating age-differences in either the slope or intercept of the function.

Inspection of the resulting model parameters (Table 3) revealed that only the main-effect for stimulus (OR $= 1.11$; $p < .001$) and the interaction between *age* and *stimulus intensity* was significant (OR $= 1.07$; $p = .003$). The main effect for *stimulus intensity* showed that participants were more likely to report that pinpricks were painful when these had high mN. The larger the effect for *stimulus intensity* the steeper the slope of the response curve. This may be interpreted as better discrimination between different stimulus intensities. Similarly the interaction between *age* and *stimulus intensity* shows that the slope of the response curve is steeper in adolescents compared to children. This may be interpreted as a better discrimination in adolescents. Absence of the main-effect for *age* indicates that no differences in response bias were present.

## DISCUSSION

The development and use of the QST protocol of the DFNS has revived interest in the methods used to collect these responses (*Rolke et al., 2006*). At the same time, methods to analyze participants' responses in such paradigms have not followed suit. Most studies in pain research rely on thresholds to describe participants' responses to painful and non-painful stimuli. In contrast, many fields have adopted psychophysical methods, specifically SDT, to describe the performance of sensory systems and, more generally, the performance of diagnostic systems (*Swets, 1988*).

The analysis of MPS data in terms of SDT may also link pain research to contemporary psychological theories of chronic pain and neuroscience. For example, the concept of interoception (*Craig, 2002*), i.e., the ability to correctly perceive internal states, has been investigated using heartbeat-detection tasks that are based in SDT (*Barrett et al., 2004*). It may

be interesting to contrast these non-nociceptive forms of interceptive awareness to data from nociceptive stimulation. SDT is widely used in cognitive neuroscience to describe non-painful sensory processes. For example, a recent study attempted to link the SDT-parameters to activations in specific brain areas (*Reckless et al., 0000*). Specifically, these authors found that shifts in response bias were associated with activation in the left dorsolateral prefrontal cortex, which also plays a key role in the modulation of pain (*Lorenz, Minoshima & Casey, 2003*). Combining the methods afforded by modern neuroscience and data analysis may yield important new insights into the nature of acute and chronic pain.

### Limitations

Also two important limitations should be kept in mind. First, applying SDT to pain-ratings requires that the data are collected using the *method of constant stimuli*, i.e., many different responses to various stimuli in a pseudorandom order need to be recorded. In addition to the disadvantages that are described above, this form of stimulation is only feasible with stimuli that can be presented at different intensities. For modalities such as heat it is much harder to present various intensities in random order. In order to present heat in random order, one would have to lift the thermode after each stimulation, wait until the temperature changes and then re-attach the thermode. Second, a major problem concerns the interpretation of the resulting model effects. As described in the introduction the early differentiation between factor that influence only psychological vs. neurological factors is not adequate (*Rollman, 1976*; *Coppola & Gracely, 1983*; *Gracely, 2006*). At present we believe it is best to use these parameters descriptively without assuming that the observed differences are due to either "psychologic" or "neurologic" mechanisms. Further research is needed to elucidate the underlying mechanisms that give rise to the observed differences and it is very likely that several reasons exist that result in a lower sensitivity or response bias. Given the wide use of the QST-protocol we are optimistic that such knowledge will be accumulated over a short period of time.

### CONCLUSIONS

As in all domains of research, progress in pain research is often associated with new methods for data collection or analysis. Although we presented an alternative to thresholds, we would like to stress that we believe that measuring thresholds using the methods of limits is valuable for clinical practice, especially when the effects under study are large and faking is not expected (*Gracely, 2006*). The approach that we advocate in this paper is a significant extension from earlier work suggesting the use of SDT in pain-research. First, we do not believe that it is possible to uphold the dogma that the response criterion is only influenced by psychological factors while the discrimination parameter is only influenced by physiological factors (*Coppola & Gracely, 1983*). Second, we believe that the use of standardized stimulation protocols is a necessary precondition to develop a robust research base into the factors that affect the SDT parameters. Third, we highlight the utility of multilevel models to estimate continuous data simultaneously on the individual and group-level. We hope that the methods presented in the present paper motivate others to utilize these methods in their research.

### Funding

The authors declare there was no funding for this work.

### Competing Interests

The authors declare there are no competing interests.

### Author Contributions

- Gerrit Hirschfeld conceived and designed the experiments, analyzed the data, contributed reagents/materials/analysis tools, wrote the paper, prepared figures and/or tables, reviewed drafts of the paper.
- Markus R. Blankenburg conceived and designed the experiments, performed the experiments, contributed reagents/materials/analysis tools, reviewed drafts of the paper.
- Moritz Süß analyzed the data, contributed reagents/materials/analysis tools, wrote the paper, prepared figures and/or tables, reviewed drafts of the paper.
- Boris Zernikow conceived and designed the experiments, contributed reagents/materials/analysis tools, wrote the paper, reviewed drafts of the paper.

### Human Ethics

The following information was supplied relating to ethical approvals (i.e., approving body and any reference numbers):

The study was approved by the Ethics Committee of the Witten/Herdecke University (92/2007) and encompassed written informed consent by children and their guardians.

### Data Availability

German data-protection laws only allow sharing research data within the participating institutions after participants gave informed consent to the storing and processing of data to a particular use. Because we did not anticipate the recent trend to share raw data when we designed the consent form and applied for ethics committee, we did not ask participants for the allowance to share the raw data. Posting this data online would constitute a violation of data protection laws.

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
