# Peer review of "Overcoming pain thresholds with multilevel models—an example using quantitative sensory testing (QST) data"

_PeerJ, doi:10.7717/peerj.1335_

## Round 0.1 · original submission · Major Revisions

Dear Authors, Please proceed to revise the manuscript accordingly as per comments of the two peer reviewers and re-submit as soon as possible. Thanking You.

Reviewer 1 ·

Basic reporting

The authors present the basic concept of signal detection theory (SDT) as a method of investigation of differences in somatosensory function. The authors introduce SDT using data from the mechanical pain sensitivity test of the quantitative sensory testing protocol as an example. The manuscript adhere to PeerJ policies.

Experimental design

The manuscript presents an interesting method of multilevel models to estimate continuous data simultaneously on the individual and group-level. This method could be interesting for the field of clinical research and clinical practice. The manuscript is well and comprehensively written. The questions posted by the authors are well defined. The methodology is clearly explained. Data presentation is clear and includes appropriate statistical methods.

Validity of the findings

No Comments

Additional comments

1. Introduction:
I recommend that the authors add a short description about QST.
Example: QST consists of seven tests measuring 13 parameters according to the QST protocol of the German Research Network on Neuropathic Pain (DFNS). It is possible to quantify the performance of human somatosensory nervous system in subjects and patients with the complete protocol. Using this method, properties of nociceptive and non-nociceptive submodalities of different groups of afferent nerve fibres and central pathways can be determined. (Mücke et al. 2014)

1.1. Thresholds
If you introduce the methods of limits and levels-method, further advantages and disadvantages should be mentioned:
To determine the sensitivity of a patient or a test subject compared to the defined test stimuli under the QST, perception and pain thresholds can be quantified. Also available are "level" and "limit" test methods - with varying advantages and disadvantages.
The "level method" entails a sensory test method, whereby stimulation is provided repetitively below and above the perception or pain thresholds. After application of the test stimuli, the subjects are asked about the perception or painfulness of the stimulus, specifically whether they are perceived as painful or not. The threshold determination is based on the stimulus intensity at which 50 % of stimuli are recognised. The disadvantage of this method is precisely the long study period required to determine a threshold; furthermore, numerous repeated measurements in determining pain stimuli can also lead to the development of sensitisation phenomena just below or above the threshold.
Another psychophysical method is the "limits" method. As part of this process, the perception and pain thresholds are recognised as the first identified stimulus under increasing stimulus intensities. In contrast to the "level" method, the "limit" method overrates the actual threshold, since the tested threshold includes a reaction time artefact. The subject has yet to give feedback after reaching the threshold, while the stimulus intensity keeps increasing further together with the reaction time. The advantage of this method is the short investigation period until the threshold determination is reached. (Mücke et al. 2014)

2. References:
Some more recent studies about QST and data analyses should be cited (see references 1-4).
List of references the authors should consider to be cited [1–4]:
1. Maier C, Baron R, Tölle TR, Binder A, Birbaumer N, Birklein F, Gierthmühlen J, Flor H, Geber C, Huge V, Krumova EK, Landwehrmeyer GB, Magerl W, Maihöfner C, Richter H, Rolke R, Scherens A, Schwarz A, Sommer C, Tronnier V, Üçeyler N, Valet M, Wasner G, Treede R-D: Quantitative sensory testing in the German Research Network on Neuropathic Pain (DFNS): Somatosensory abnormalities in 1236 patients with different neuropathic pain syndromes. PAIN 2010, 150:439–450.
2. Pfau DB, Krumova EK, Treede R-D, Baron R, Toelle T, Birklein F, Eich W, Geber C, Gerhardt A, Weiss T, Magerl W, Maier C: Quantitative sensory testing in the German Research Network on Neuropathic Pain (DFNS): reference data for the trunk and application in patients with chronic postherpetic neuralgia. Pain 2014, 155:1002–1015.
3. Magerl W, Krumova EK, Baron R, Tölle T, Treede R-D, Maier C: Reference data for quantitative sensory testing (QST): refined stratification for age and a novel method for statistical comparison of group data. Pain 2010, 151:598–605.
4. Mücke M, Cuhls H, Radbruch L, Baron R, Maier C, Tölle T, Treede R-D, Rolke R: Quantitative sensorische Testung. Schmerz 2014, 28:635–648.

3. Limitations:
Limitations of SDT should be explicitly explained.

4. Figures:
Figure 2: The representation of Figure 2 should be reconsidered. The panels are too small and not readable, therefore not easy understandable.

Some minor points:
1. Some minor typos and grammar should be corrected.
2. Keywords: Quantiative sensory testing (QST) should be added, since SDT has an impact on QST.
3. Please introduce abbreviations such as GLM.
4. Special terms e.g. “d’ and C parameters” should be explained for laypersons (postgraduate, students etc.), which could also be interested in this paper.

Reviewer 2 ·

Basic reporting

No comments

Experimental design

A more detailed information about the methods (e.g. MPS) would be helpful for the user. E.g. the authors state that they have analyzed data from 172 children (p. 8). They state also: "We analyzed the individual responses to 15.480 responses of the children" (p.9) The link is unclear for the reader.

Validity of the findings

No comments

Additional comments

The methods, the results and figure 2 would benefit if they were more detailed and clearly explained for the readers without profound expertise on statistical methods.

---

## Round 0.2 · Minor Revisions

Dear Authors, Once you can complete the hanging incomplete sentence as requested by Peer Review 2, the manuscript can be processed further

Reviewer 1 ·

Basic reporting

The authors present the basic concept of signal detection theory (SDT) as a method of investigation of differences in somatosensory function. The authors introduce SDT using data from the mechanical pain sensitivity test of the quantitative sensory testing protocol as an example. The manuscript adhere to PeerJ policies.

Experimental design

The manuscript presents an interesting method of multilevel models to estimate continuous data simultaneously on the individual and group-level. This method could be interesting for the field of clinical research and clinical practice. The manuscript is well and comprehensively written. The questions posted by the authors are well defined. The methodology is clearly explained. Data presentation is clear and includes appropriate statistical methods.

Validity of the findings

No Comments

Additional comments

There are still some minor typing errors in the manuscript, but all my questions and comments were answered satisfactorily.

Reviewer 2 ·

Basic reporting

The manuscript has been significantly improved.

Experimental design

No Comments

Validity of the findings

No Comments

Additional comments

p. 16, last raw: The sentence seems to end abrupt: "At present we believe it is best to use these parameters descriptively without" Without what? Please add the missing words.

---

## Round 0.3 · accepted · Accept

Dear Authors,

Thank You for your hard work in revising the manuscript in accordance to the comments of the peer reviewers. We are happy to accept the manuscript and will process it further so as it can be publish in the shortest possible time.